# Risk Factors for Complications Following Staged Alveolar Ridge Augmentation and Dental Implantation: A Retrospective Evaluation of 151 Cases with Allogeneic and 70 Cases with Autogenous Bone Blocks

**DOI:** 10.3390/jcm12010006

**Published:** 2022-12-20

**Authors:** Frank R. Kloss, Peer W. Kämmerer, Anita Kloss-Brandstätter

**Affiliations:** 1Oral- and Maxillofacial Surgeon, Private Clinic for Oral- and Maxillofacial Surgery, Kärntnerstraße 62, 9900 Lienz, Austria; 2Department of Oral and Maxillofacial Surgery, University Medical Centre Mainz, Langenbeckstraße 1, 55131 Mainz, Germany; 3Department of Engineering & IT, Carinthia University of Applied Sciences, Europastraße 4, 9524 Villach, Austria

**Keywords:** alveolar ridge augmentation, bone allograft, allogeneic, bone autograft, oral bone grafting, dental implant, retrospective evaluation

## Abstract

Purpose: the aim of this study was to identify potential risk factors favoring complications by assessing the number and types of complications associated with allogeneic or autogenous bone blocks applied as onlay grafts for alveolar ridge augmentation prior to implantation. Methods: A retrospective chart review on the success of 151 allogeneic and 70 autogenous bone blocks in a cohort of 164 consecutive patients, who were treated over a period of 6 years by the same surgeon, was conducted. Statistical conclusions were based on ROC curves and multiple logistic regression models. Results: Complications were observed more frequently with autogenous bone blocks (14 out of 70 cases; 20%) compared to allogeneic bone blocks (12 out of 151 cases; 7.9%; *p* = 0.013). However, these complications were minor and did not impact the successful dental rehabilitation. In a multiple logistic regression model, the risk of a complication was increased by the use of an autogenous bone block (OR = 3.2; *p* = 0.027), smoking (OR = 4.8; *p* = 0.007), vertical augmentation above a threshold of 2.55 mm (OR = 5.0; *p* = 0.002), and over-contouring (OR = 15.3; *p* < 0.001). Conclusions: Overall, the complication rate of ridge augmentations carried out with autogenous or allogeneic bone blocks was low. Despite previous recommendations, over-contouring and a vertical augmentation above a threshold of 2.55 mm should be avoided.

## 1. Background

Alveolar ridge defects resulting from tooth loss may compromise dental restoration with implants [1], so augmentation of the deficient ridge ahead of implant placement to restore the required bone volumes is a common practice [2,3,4]. Autogenous bone blocks have been successfully applied in preimplant surgery with excellent clinical outcomes and high implant survival rates for decades [5,6,7,8]. Nonetheless, harvesting of autogenous bone was associated both with donor site morbidity that may severely increase patient burden and with the risk of nerve damage, despite the establishment of widely safe and predictable bone-harvesting procedures and donor sites [9,10,11,12]. As a consequence, allogeneic bone blocks have been considered an increasingly popular grafting material, as several studies emphasized the feasibility of achieving results comparable to those of autogenous bone blocks while omitting the drawbacks of bone harvesting [13,14].

A recently published systematic review testing the hypothesis of no difference in implant treatment outcome after horizontal ridge augmentation with allogeneic as compared to autogenous bone blocks [15] found no significant differences and hence was in conformity with a comparative short-term study recently published by our group [16]. However, there has been an ongoing debate regarding the increased risk of complications associated with allogeneic bone blocks [15,17], whereby cohort sizes in previous studies reporting complication rates were rather small (*n* = 10 [18]; *n* = 12 [19]; *n* = 14 [20]; *n* = 15 [21]; *n* = 16 [22]; *n* = 19 [23]; *n* = 20 [24,25]; *n* = 40 [26]; *n* = 58 [27]; *n* = 6 [17]).

The aim of this retrospective study was to identify potential risk factors for complications in staged alveolar ridge augmentation and dental implantation. Therefore, the number and type of complications observed with 151 allogeneic and 70 autogenous bone blocks used for alveolar ridge augmentation prior to dental implantation were assessed over a period of six years in a cohort of 164 patients from a private clinic for oral- and maxillofacial surgery.

## 2. Materials and Methods

### 2.1. Study Design and Setting

The charts of all patients, who received allogeneic or autogenous bone blocks for the purpose of alveolar ridge augmentation and were consecutively treated between March 2013 and March 2019 in a private clinic for oral- and maxillofacial surgery by one surgeon in Eastern Tyrol, were consecutively included in this retrospective chart review.

### 2.2. Ethics Statement

The study was approved by the Ethics Committee of the State Medical Association of Rheinland-Pfalz (approval no. 2022-16445). The study was performed in accordance with the STROBE guidelines. The authors of this manuscript confirmed that the recognized standards of the Declaration of Helsinki had been followed. All subjects provided their informed consent prior to inclusion in the study.

### 2.3. Participants

A total of 164 consecutive patients with one or several missing teeth and insufficient bone quantity for direct implantation, who underwent allogeneic or autogenous bone block augmentation procedures, were enrolled into this retrospective study. At baseline, all sites were fully healed. All patients were fully informed about the surgical procedures and treatment alternatives. Only patients with single-tooth gaps or small edentulous gaps were included but not complete edentulous patients.

Exclusion criteria consisted of a history of radiotherapy in the head and neck region, systemic disease that would contraindicate oral surgery, uncontrolled periodontal disease, bruxism, pregnancy, and psychiatric problems. All other patients, also those suffering from diabetes and those receiving oral bisphosphonates, who were presenting with a bone atrophy of the alveolar ridge in the predominantly horizontal and/or vertical plane as identified by cone beam computed tomography (CBCT) para-axial reconstruction images, were enrolled in this retrospective evaluation.

Every patient was subjected to three-dimensional X-ray diagnostics (CBCT), followed by computer-aided planning of the augmentation and subsequent implantation. In total, four CBCTs were recorded for each patient, one before treatment, one directly after augmentation, one after six months of healing, and one after 12 months. The defect sizes were classified according to the ITI-treatment guide 7 categories [28].

### 2.4. Allogeneic Bone Blocks

All patients received cancellous freeze-dried allogeneic bone blocks (maxgraft^®^, botiss biomaterials GmbH, Zossen, Germany) made from bone of explanted femoral heads provided by living donors subjected to hip arthroplasty treatment. Following an assessment of the donor’s health status, the femoral heads were forwarded to a tissue bank where they are split, cleaned, degreased in an ultrasonic bath, and wet-chemically purified. Subsequent lyophilization of the bone material ensured long-term storability at room temperature, while gamma-irradiation was applied for the terminal sterilization [29,30,31].

### 2.5. Autogenous Bone Blocks

Bone blocks were harvested in most cases from the external oblique line of the mandible with piezo surgery. Other donor sites included the iliac crest and the tuber maxillae. At the donor site, a paramarginal incision was made in the molar area if teeth were present or on top of the alveolar crest in the case of an edentulous ridge. A full-thickness flap was elevated, exposing the external oblique ridge and the lateral aspect of the ramus as well as the lateral aspect of the mandibular body.

The osteotomy cuts were prepared with a piezoelectric instrument (Piezomed, W&H Dentalwerk, Bürmoos, Austria). The block was removed with a straight, thin chisel without the need for hammering. The flap was sutured using single sutures. Resorba RESOLON^®^ blue monofil (USP 4-0) and Cytoplast™ Suture (USP 4-0) were used as suture materials.

The autogenous bone blocks were adapted to the defect site and then grafted in combination with autogenous bone particles scraped from the same lamina [32,33].

### 2.6. Surgical Procedure

The pre-operative clinical examinations, the surgical procedure, and the post-operative care were carried out as previously described [16]. Midline crestal incision was used as well as a mobilization of the mucosa by perforating the periosteum. Briefly, the native bone was decorticated with drills to induce bleeding and promote vascularization of the grafting material [34]. The allografts were obtained sterile from the double pouch, adapted in accordance with the defect morphology, and fixated onto the host bone with two 1.5 mm osteosynthesis screws (Komet Dental, Lemgo, Germany). If necessary, small void spaces were filled with a bovine bone substitute material (either Endobon^®^, Biomet 3i LLC, Munich, Germany, or cerabone^®^, botiss biomaterials GmbH, Zossen, Germany). The surgical site was protected with a resorbable barrier membrane made from porcine pericardium (Jason^®^ membrane, botiss biomaterials GmbH, Zossen, Germany). The membrane was not fixed with pins or sutures but placed underneath the periosteum of the oral aspect and superimposed the allogeneic bone block towards the vestibular site. The intervention was carried out under local anesthesia. Routine post-operative care included administration of amoxicillin and clavulanic acid (625 mg, administered orally, three times a day for 4 days), ibuprofen (600 mg, administered orally, every 6 h as needed), and mouthwashes (0.2% chlorhexidine, three times daily for 7 days). The patients were recalled at monthly intervals for a period of six months to detect possible complications, such as infection, pain, discomfort, graft exposure, and graft mobility.

### 2.7. Implantation

The implants were placed after six months of healing as described previously [16]. Fixation screws were removed, and the graft stability was assessed. Every patient received one or two titanium implants per augmented region. The inserted implants were from Straumann (*n* = 194; Type SLActive^®^; Straumann Holding AG, Basel, Switzerland), bredent (*n* = 19; blueSKY; bredent medical GmbH & Co. KG, Senden, Germany), Camlog (*n* = 3, CAMLOG Biotechnologies GmbH, Basel, Switzerland), Astra Tech (*n* = 1, Dentsply Sirona Austria GmbH, Vienna, Austria), Ankylos (*n* = 3, Dentsply Sirona Austria GmbH, Vienna, Austria), and Medentika (*n* = 1, Straumann Holding AG, Basel, Switzerland).

### 2.8. Evaluation and Variables

The following outcome variables were collected:
Implant survival.Occurrence and type of a complication.
○Wound dehiscence.○Partial loss of allogeneic block.○Total loss of allogeneic block.○Loss of implant.○Infection.


The following predictors and potential confounders were extracted from the patient records:Implant location (upper or lower jaw).Gender.Age.SmokingMedication.ITI defect classification.Defect size (single-tooth gap or several missing teeth).Height of the alveolar ridge before augmentation (mm).Height of the alveolar ridge after augmentation (mm).Width of the alveolar ridge before augmentation (mm).Width of the alveolar ridge after augmentation (mm).Vertical augmentation (mm).Horizontal augmentation (mm).Use of a barrier membrane.Type of granular grafting material (cerabone or Endobon).Type of implant.Length of implant.Diameter of implant.Prosthetic restoration (crown; fixed partial denture; apex locator; bar prosthesis; telescope prosthesis).

Over-contouring was defined as an over-augmentation of the alveolar process beyond the extent of the adjacent bone level (Figure 1). The reason for over-augmenting the alveolar process is often an anticipated shrinkage of the bone substitution material during the healing phase. Two patients were selected as examples to illustrate the term “over-contouring”. Patient 1 showed a Class III defect in the maxilla, and the alveolar ridge augmentation performed for correction was over-contoured (Figure 2). Patient 2 also showed a Class III defect in the maxilla, but the alveolar ridge augmentation performed for correction remained at the level of the surrounding bone (Figure 3).

Patient compliance was categorized according to following criteria:Good compliance: compliance with all required control appointments and behavioral measures (oral hygiene, rinsing, pause of wearing prostheses)Moderate compliance: missing one control appointment or two control appointments with catching up at a later point in time; failure to comply with behavioral measures (oral hygiene and rinsing)Poor compliance: missing two or more control appointments or non-compliance with any prosthesis absence if required.

### 2.9. Study Size and Potential Sources of Bias

In order to eliminate a potential selection bias, all consecutive patients who were treated with allogeneic or autogenous bone blocks for alveolar ridge defects between 2013 and 2019 were included in this retrospective evaluation. Within the observational period of six years, 164 patients were treated.

### 2.10. Statistics

Statistical analyses were performed with IBM SPSS (version 27; International Business Machines Corp., Armonk, NY, USA) and with RStudio, applying the libraries *mosaic*, *devtools*, *ggiraph*, *ggiraphExtra*, *moonBook*, and *plyr*. For descriptive statistics of quantitative variables, mean values and standard deviations were calculated. The data set was complete, and there were no missing data. Pearson’s chi-squared test was applied to sets of unpaired categorical data to evaluate the likelihood that any observed difference between the sets was due to chance. Fisher’s exact test was used where sample sizes were small. An independent sample *t*-test was used when two separate sets of independent and identically distributed samples were obtained, and their population means were compared to each other. A Mann–Whitney U test was used as a nonparametric test of the null hypothesis that, for randomly selected values X and Y from two populations, the probability of X being greater than Y was equal to the probability of Y being greater than X (e.g., for comparing the lengths of the implants between the two treatment groups).

To investigate the influence of the amount of vertical augmentation on the occurrence of complications, an ROC curve was created to represent the highest Youden’s index. This is calculated from *J = max_i_* [*sensitivity*(*i*) *+ specificity*(*i*) *−* 1] [35,36,37]. Using Youden’s index, a cut-off value was determined for the variable “vertical augmentation”, and with that, the variable was dichotomized (low vertical augmentation versus pronounced vertical augmentation).

Using multiple logistic regression, the influence of several independent variables on the occurrence of dehiscence or partial loss of the allogeneic bone block was examined as binary variables. The regression model was used to determine the *p*-values, the relative risk (odds ratio), and the associated 95% confidence interval of the individual independent variables. The Nagelkerke pseudo-R-squared coefficient of determination was calculated as the equivalent of the explained variance.

## 3. Results

### 3.1. Study Population

The demographic characteristics of the study population are summarized in Table 1. Gender was distributed evenly between the two study groups (chi-squared test; *p* = 0.504). Patients receiving autogenous bone blocks for alveolar ridge augmentation were on average 6 years younger than patients receiving allogeneic bone blocks (*t*-Test; *p* = 0.002). Patients from both study groups had a mean follow-up time of 3.5 years with no difference in the follow-up time between the two study groups.

The bone defects were classified as type II (non-self-containing dehiscence defect with bone eminences next to adjacent teeth) for 131 patients, as type III (combined horizontal and vertical defect) for 64 patients, and as type IV (through-and-through defect) for 26 patients according to the ITI treatment guide categories [38]. There was a significant difference in the distribution of bone defects between the two study groups, with dominance of type-II defects in the autogenous bone graft group (chi-squared test; *p* = 0.043). Accordingly, there was a significant association between the size of the defect and the treatment group: in the autogenous study group, the single-tooth gaps dominated, while in the allogeneic study group, defects with several missing teeth were predominant (chi-squared test; *p* = 0.002). There was no difference in smoking, patient compliance, and over-contouring between the two study groups.

Patients with a single-tooth gap were on average 47.3 ± 14.0 years old, while patients with several missing teeth were on average 51.1 ± 13.6 years old, with the difference being statistically significant (*t*-test; *p* = 0.044). This is one possible explanation for the fact that patients receiving autogenous bone blocks were significantly younger than patients receiving allogeneic bone blocks: larger defects required a treatment with a larger bone graft, which could not have been harvested from the external oblique line of the mandible.

### 3.2. Ridge Augmentation, Implantation, and Dental Rehabilitation

In total, 221 bone blocks (151 allografts and 70 autografts) were used for augmentation of the alveolar process in 164 patients. One hundred and thirteen (68.9%), forty-six (28.1%), and four (2.4%) patients received one, two, or three bone blocks, respectively. One patient (0.6%) received four bone blocks. The bigger part of augmentations was carried out in the maxilla (*n* = 132; 59.7%). An interesting observation was that while in the autogenous treatment group, augmentations were balanced between maxilla and mandible, in the allogeneic treatment group most augmentations (64.9%) were performed in the maxilla (chi-squared test; *p* = 0.027). Over-contouring was applied in 48 cases (21.7%), with no difference between the two treatment groups. In 125 out of 221 augmentations (56.6%), the treatment aimed at compensating several missing teeth, while in 96 augmentations (43.4%) the augmentation was performed for restoring a single-tooth gap. The length of the inserted implants ranged from 6 mm to 12 mm (median: 10 mm). In the autogenous treatment group, the median length of the implants was 10 mm, while in the allogeneic treatment group, the median length of the implants was 9 mm (Mann–Whitney test; *p* = 0.001). The diameter of the inserted implants ranged between 2.9 mm and 4.8 mm (3.9 ± 0.4 mm), with no difference between the two treatment groups. Crowns were installed at more than the half (*n* = 120; 54.5%) of all augmentation sites (autogenous: *n* = 52; allogeneic: *n* = 68), while 91 (41.4%) augmentations were associated with final restorations by fixed partial dentures (autogenous: *n* = 18; allogeneic: *n* = 73). Five allogeneic bone blocks were applied for supporting the telescope prosthesis, three allogeneic bone blocks were used for implants that served as substructure for apex locator prosthesis, and one allograft was attached to insert an implant for a bar prosthesis.

### 3.3. Complications Observed with Alveolar Ridge Augmentation

Overall, in 195 out of 221 augmentations (88.2%), no complications were observed (Table 2). While only 56 out of 70 autogenous augmentations (80.0%) were complication-free, 139 out of 151 allogeneic augmentations (92.1%) were without any complications. Wound dehiscence accompanied by partial loss of the allogeneic block occurred in nine augmentative procedures (4.1%) and in three cases without further negative implications (1.4%). Total graft loss was noted in two patients (0.8%) due to wound dehiscence, in one patient due to deep infection, and without prior complications in another patient (0.7%), respectively. Additionally, eight patients (3.6%) presented with partial graft loss without former complications. One implant was lost two months after implantation in the allogeneic group, but a subsequent implantation was successful.

Overall, the occurrence of a complication was significantly associated with the type of the bone block (chi-squared test; *p* = 0.013), with autograft being more prone to complications than allografts (Table 2). Indeed, a partial loss of the bone block was significantly more frequently observed after autogenous augmentation of the alveolar ridge than after allogeneic augmentation (Fisher’s exact test; *p* = 0.001). All other complications showed equal frequencies within the two study groups.

### 3.4. Risk Factors for Complications with Alveolar Ridge Augmentation

As stated above, the occurrence of a complication was significantly associated with the type of the bone block (chi-squared test; *p* = 0.013), with autograft being more prone to complications than allografts (Table 3). Patients suffering from a complication after alveolar ridge augmentation were significantly younger than patients experiencing no complications (*t*-test; *p* = 0.007).

The occurrence of a complication was associated with smoking (chi-squared test; *p* = 0.018), poor patient compliance (Fisher’s exact test; *p* < 0.001), the ITI defect classification (Fisher’s exact test; *p* = 0.024), and over-contouring (Fisher’s exact test; *p* < 0.001). The other covariates (location in the maxilla or mandible, medication, defect size, application of granular bovine bone grafting material, and the extent of the horizontal augmentation) were not associated with the manifestation of a complication (Table 3).

The most intriguing observation was that the extent of the vertical augmentation was highly significantly associated with the occurrence of a complication. Augmentations in which a complication occurred had an average vertical bone augmentation height of 2.9 mm, while augmentations without complications had an average vertical bone augmentation height of 1.0 mm (*t*-test; *p* < 0.001).

In order to establish a threshold above which vertical augmentation could become more problematic, Youden’s index was calculated based on the ROC curve of the amount of vertical augmentation (Figure 4). Based on 221 data points on vertical alveolar ridge augmentation, Youden’s index was inferred as 2.55 mm, indicating that vertical augmentations below a height of 2.55 mm were associated with a low risk of complications, while vertical augmentations above a height of 2.55 mm were associated with a high risk of complications. Therefore, the metric variable “height of the vertical augmentation” was transformed into a binary classifier system with a discrimination threshold of 2.55 mm. This new binary variable was then subjected to an association analysis with the occurrence of a complication, and the association was highly significant (Table 3; chi-squared test; *p* < 0.001). If the vertical augmentation height was below the threshold, complications occurred in only 6.5% of augmentations. If the vertical augmentation height was above the threshold, complications occurred in 29.4% of augmentations.

Since the extent of necessary reconstruction of the alveolar process is strongly dependent on the size of the underlying bone defect, the association analysis between the extent of vertical augmentation and the occurrence of complications was stratified according to the ITI defect class (Figure 5). As expected, in low defect classes (type I and II), only few vertical augmentations lying above the critical value of 2.55 mm were performed (7 out of 131 augmentations; 5.3%). There was no association with the occurrence of a complication (*p* > 0.05). However, in higher defect classes (type III and IV), vertical augmentations lying above the threshold value of 2.55 mm were performed in half of the cases (44 of 90 augmentations; 48.9%). There was a significant association with the occurrence of a complication (*p* = 0.011).

A detailed analysis on the association between over-contouring and the occurrence of a complication can be found in Figure 5. Looking first at autogenous augmentations, it is noticeable that complications only occurred when there was over-contouring of the alveolar ridge. Conversely, no complications occurred at all if the augmentation did not extend above the natural level of the alveolar ridge. Over-contouring with autogenous bone led to a complication in 78% of cases. The picture was different for allogeneic augmentations: Complications could only be attributed to over-contouring in 58% of the cases. However, over-contouring with allogeneic bone blocks led to a complication in only 17% of cases.

### 3.5. Multiple Logistic Regression Analyses

To disentangle the effects of the diverse significant variables from Table 3 on the outcome variable “complication”, a multiple logistic regression analysis was performed (Table 4). The following variables were found to be significantly associated with complications in univariate statistical analysis and were therefore primarily included as covariates in the multiple logistic regression model: bone block material (autogenous or allogeneic), age, smoking, ITI defect classification, vertical augmentation as binary classifier, patient compliance, and over-contouring. However, in the multiple regression model, the variables patient compliance (*p* = 0.367), age (*p* = 0.169), and ITI defect classification (*p* = 0.148) were no longer statistically significant and were therefore removed from the model. The application of autogenous bone blocks, smoking, over-contouring, and a vertical augmentation height above the threshold level of 2.55 mm remained the only variables to be significantly associated with the risk of a complication (Table 4).

The application of an autogenous bone block instead of an allogeneic bone block raised the risk of a complication by a factor 3.2. Smokers had a 4.8 times higher risk of suffering from a complication than non-smokers. If the height of the vertical augmentation was above the threshold of 2.55, the risk of a complication increased by a factor of five. Over-contouring of the alveolar ridge was associated with an increased risk of complication by a factor of 15.3. With this statistical model, 93.2% of cases could be predicted correctly (Table 5). This indicated a good explanatory power of the statistical model, which is also supported by a Nagelkerke pseudo-R-square value of 0.453.

A detailed analysis of the reasons for the lack of association between patient compliance and the occurrence of a complication in the multiple logistic regression model revealed that smoking and patient compliance were significantly associated with each other (Pearson’s chi-squared test; *p* = 0.003). The patient compliance was significantly lower in smokers compared to non-smokers (Table 6).

Figure 6 shows the results of the logistic regression analysis for the probability of a complication. In the overall group, it can be clearly seen that over-contouring entails an increased risk of a complication with an increasing height of the vertical augmentation. Only the pure height of the vertical augmentation also steadily increases the risk of a complication, but as soon as the height of the bone augmentation exceeds the level of the surrounding, natural alveolar ridge, the risk of a complication multiplies. If the analysis is limited to the autogenous bone blocks, this association becomes even more apparent. Here, it can be seen that with increasing vertical augmentation, the risk of a complication does not increase at all as long as the augmentation remains below the level of the surrounding, natural alveolar ridge. Over-contouring, however, is little tolerated. The course of the light blue curve, i.e., the risk curve in the presence of over-contouring, once again underlines the findings of the ROC analysis: from a vertical height gain of 2.55 mm with simultaneous over-contouring, the risk of a complication is already 90%.

If the analysis is restricted to the allogeneic bone blocks, it becomes apparent that the risk of a complication increases with increasing vertical augmentation height but never exceeds a probability of 50%. In addition, the course of the light blue curve (over-contouring) is similar to the course of the dark blue curve (no over-contouring), so that it can be concluded that over-contouring is more likely to be tolerated with allogeneic bone than with autogenous bone.

## 4. Discussion

This retrospective study of patient charts evaluated the survival and complications observed in 151 allogeneic and 70 autogenous bone blocks that were used for alveolar ridge augmentation prior to dental implantation in a cohort of 164 consecutive patients treated by the same surgeon over a period of 6 years. To the best of our knowledge, this was the first retrospective clinical study analyzing all patients treated for alveolar ridge defects with either grafting material within a period of six years, which included such an extensive patient collective. It is important to point out that the patients had chosen autogenous or allogenous bone grafting themselves. All patients were informed by the oral surgeon about both bone grafting materials, but the decision for allogenic or autogenic bone for alveolar ridge augmentation was left to the patient.

The overall observed complication rate of 11.8% (26 out of 221 cases) was low and complied with findings of previous reports [13,39,40,41]. Complications were observed more frequently with autogenous bone blocks (20%) compared to allogeneic bone blocks (7.9%). A partial loss of the bone block was noted more frequently with autogenous bone blocks (18.6%) compared to allogeneic bone blocks (4.6%). No significant differences were observed in the frequencies of wound dehiscence, infection, implant loss, and total loss of the bone block. In a multiple logistic regression model, the risk of a complication was increased by the use of an autogenous bone block (OR = 3.2; *p* = 0.027), smoking (OR = 4.8; *p* = 0.007), vertical augmentation above a threshold of 2.55 mm (OR = 5.0; *p* = 0.002), and over-contouring (OR = 15.3; *p* < 0.001).

When the massively increased risk of complications of alveolar ridge augmentations due to over-contouring was considered, one may ask why over-contouring was performed at all. Even when comparing the two radiological images in Figure 2 (over-contouring) and Figure 3 (no over-contouring), it seemed questionable why over-contouring should be useful at all. However, the recommendation for over-contouring, especially with autogenous bone material, has been found in numerous literature references.

Previous authors observed that autografts were subject to excessive remodeling after surgery, especially when no barrier membranes were used [42], and found that mean graft resorption of onlay grafts harvested from the iliac crest ranged between 15% [43] and 25% [44]. Because of these observations made for autogenous bone blocks, over-contouring of the ridge with grafting material in order to ensure sufficient ridge volume for adequate support of the implant body and obviate the risk of threat exposure was suggested and established as a common procedure [9,45]. Finally, in a systematic review and meta-analysis on the fate of lateral ridge augmentations, the authors concluded that regardless of the material used for regeneration, “overcorrection of the horizontal defects should be performed to compensate for the resorption of the grafting materials” [46]. However, our previous comparative analysis on the graft shrinkage of autogenous and allogeneic bone blocks found that the mean volume reduction after 12 months was 12.5 ± 7.8% and 14.4 ± 9.8% for autogenous and allogeneic bone blocks, respectively [16]. These findings emphasized that over-contouring of the alveolar ridge when using allogeneic bone blocks for treating bone defects classified as ITI type II to type IV should preferably be omitted [38].

Another very impressing observation was that while autogenous bone reacted very sensitively to over-contouring, allogenous bone appeared to be more tolerant with regard to over-contouring. So, contrary to Draenert et al. [17], who only included six allogeneic blocks in their study and who concluded that vertical augmentations with allogeneic bone blocks should be avoided at all, our much bigger study resulted in the finding that allogeneic bone blocks were more tolerant to vertical augmentations as compared to autogenous bone blocks. However, something else was observed too: the complication rate of autogenous bone was zero as long as the augmentation did not exceed the surrounding bone material. Of course, this also meant that smaller bone blocks were harvested and the morbidity at the harvest site was lower.

Overall, there was no difference in the success rate in terms of implant survival between autogenous and allogeneic bone blocks. This was in concordance with a systematic review and meta-analysis on the success rate of implants placed in autogenous bone blocks versus allogeneic bone blocks found that implant survival ranged from 73.8% to 100% in autogenous bone blocks and from 72.8% to 100% in allogeneic bone blocks, with no significant difference between the two study groups [41].

An initially irritating observation was that while patient compliance did indeed show an influence on the occurrence of complications in the individual statistical analysis, the association was lost in the multiple logistic regression model. One reason for the lack of association between patient compliance and complications after adjustment for other covariates in the multiple regressions model was the significant association of patient compliance and smoking combined with the reduced compliance of smokers as compared to non-smokers. In fact, prior descriptive epidemiology studies showed that smokers have lower compliance rates with preventive care services than non-smokers [47]. Therefore, smoking was detrimental for the success of oral surgery in several interlaced aspects, as tobacco negatively affects wound healing on multiple levels [48], while the inferior patient compliance of smokers aggravated this situation.

A very frequently cited study on complications with allogeneic cancellous bone blocks used for augmentation of the alveolar ridge found a rate of wound dehiscence (membrane exposure) of 30.7% [49]. This extremely high value could not be confirmed. In our study, only 10% of the autogenous and 5.3% of the allogenic bone blocks were affected by wound dehiscence. A quick statistical review revealed that in 10 of the 15 cases of wound dehiscence occurring in our study, over-contouring had been performed. The association between wound dehiscence and over-contouring was highly significant (*p* < 0.001). It could be concluded that over-contouring of the alveolar ridge should be strongly discouraged to avoid membrane exposure. Since the study with the high rate of membrane exposure was published in 2010 [49], and since at that time over-contouring was recommended by default based on the experiences with autogenous bone blocks harvested from the iliac crest, it could be assumed that even then over-contouring of the alveolar ridge led to the observed complications.

The extensive number of consecutive cases (*n* = 221) combined with the detailed recording of all anamnestic characteristics and treatment modalities, which facilitated identification of over-contouring as a risk factor, are unique to the present study. Regarding the present study, our goal was to systematically report complications associated with autogenous and allogeneic bone blocks in oral preimplant restorative surgery. In our earlier systematic comparative study on the shrinkage behavior of bone blocks, there was no difference in the complication rate between allogeneic and autogenous bone blocks, though [16].

Summarized, our retrospective evaluation of 221 augmentations with 151 allogeneic and 70 autogenous bone blocks indicated that smoking, a vertical gain height of more than 2.55 mm, and over-contouring were strongly associated with the occurrence of a complication. Our previous volumetric comparative study emphasized low and nearly identical graft resorption rates of both autogenous bone blocks obtained from the external oblique line of the mandible and cancellous allogeneic bone blocks in horizontal bone augmentations [16]. Therefore, over-contouring of the alveolar process did not only seem to be redundant but might have also been detrimental for the success of alveolar ridge augmentation. Future complication analyses on allogeneic bone blocks but also on other grafting materials should further corroborate these observations by including the variables over-contouring and gain in vertical height into their list of covariates.

## 5. Conclusions

To the best of our knowledge, this was the first study clearly identifying over-contouring and extended vertical augmentation as potential major risk factors in alveolar ridge augmentation. In our investigation, vertical augmentation of more than 2.55 mm was the threshold for a possible complication. In conclusion, it is strongly recommended to refrain from over-contouring of the alveolar process during ridge augmentation.

## Figures and Tables

**Figure 1 jcm-12-00006-f001:**
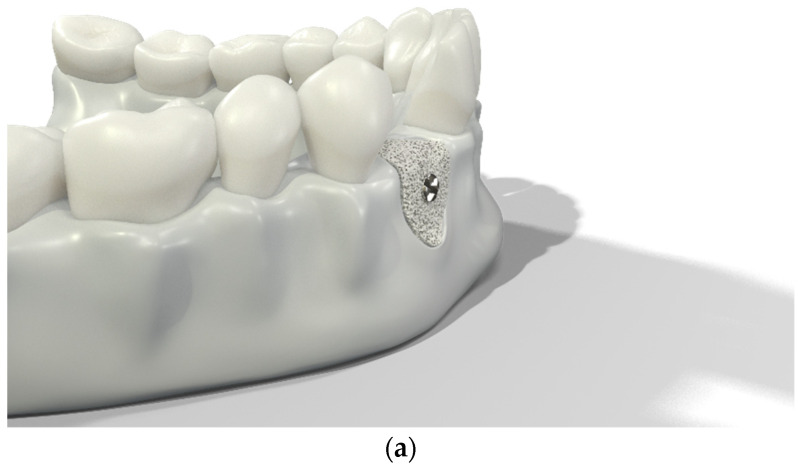
Graphical 3D models to demonstrate surgical over-contouring during augmentation of the alveolar process. (**a**) Lateral view of the mandible with a single-tooth gap on the position of the second right incisor; the bone defect was compensated without exceeding the extent of the adjacent bone level. (**b**) Lateral view of the mandible with a single-tooth gap on the position of the second right incisor; the bone defect was over-compensated by an over-augmentation of the alveolar process beyond the extent of the adjacent bone level. This is defined here as “over-contouring”. (**c**) Isometric view of the mandible with a single-tooth gap on the position of the second right incisor; the difference between appropriate augmentation (left half) and over-contouring (right half) is clearly visible. (**d**) Lateral view of the mandible with a single-tooth gap on the position of the second right incisor; the difference between appropriate augmentation (left half) and over-contouring (right half) is easily recognizable.

**Figure 2 jcm-12-00006-f002:**
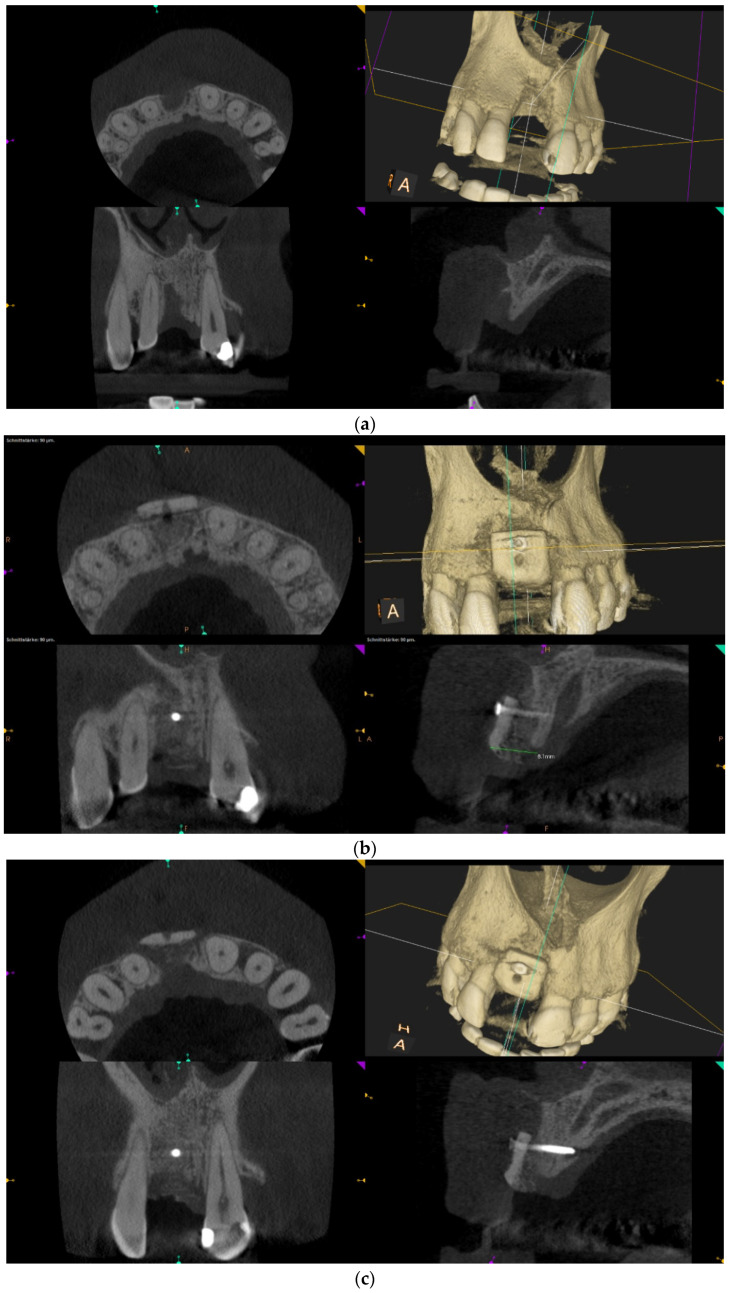
Clinical example of over-contouring. (**a**) Class III defect with pronounced vestibular deficit and minor vertical deficit. (**b**) Post-OP: Attachment of a cortico-cancellous allogeneic bone block. Vestibular over-contouring with the cortical plate and vertical over-contouring of the block can be seen. The block protrudes over the limbus alveolaris, i.e., over the bone border of the adjacent teeth. (**c**) After 5 months: the over-contoured portion of the cortical portion of the block was not resorbed, but part of the cancellous portion of the block was palatally resorbed away. (**d**) The over-contoured block penetrates the mucosa. The cortical portion is revealed.

**Figure 3 jcm-12-00006-f003:**
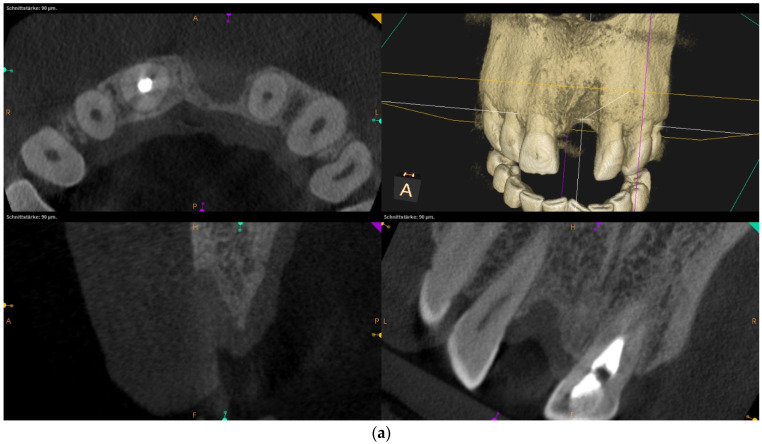
Clinical example of no over-contouring. (**a**) Class III defect with pronounced vestibular deficit and minor vertical deficit. (**b**) Attachment osteoplasty with a cancellous allogeneic bone block. The block fits into the contour of the surrounding alveolar process (within the envelope). The lining with the bovine granules is visible (vestibular opaque line). Cranially, the bone block ends at the bone border of the adjacent teeth.

**Figure 4 jcm-12-00006-f004:**
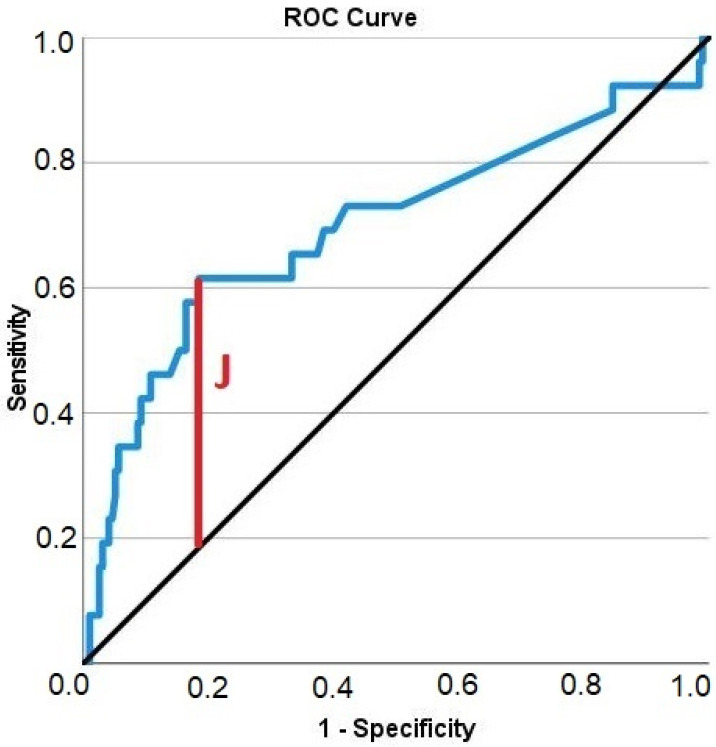
Receiver operating characteristic curve illustrating the diagnostic ability of the height of the vertical alveolar ridge augmentation in predicting complications. Solid blue: ROC curve; black diagonal line: chance level; vertical red line (J): maximum value of Youden’s index for this ROC curve.

**Figure 5 jcm-12-00006-f005:**
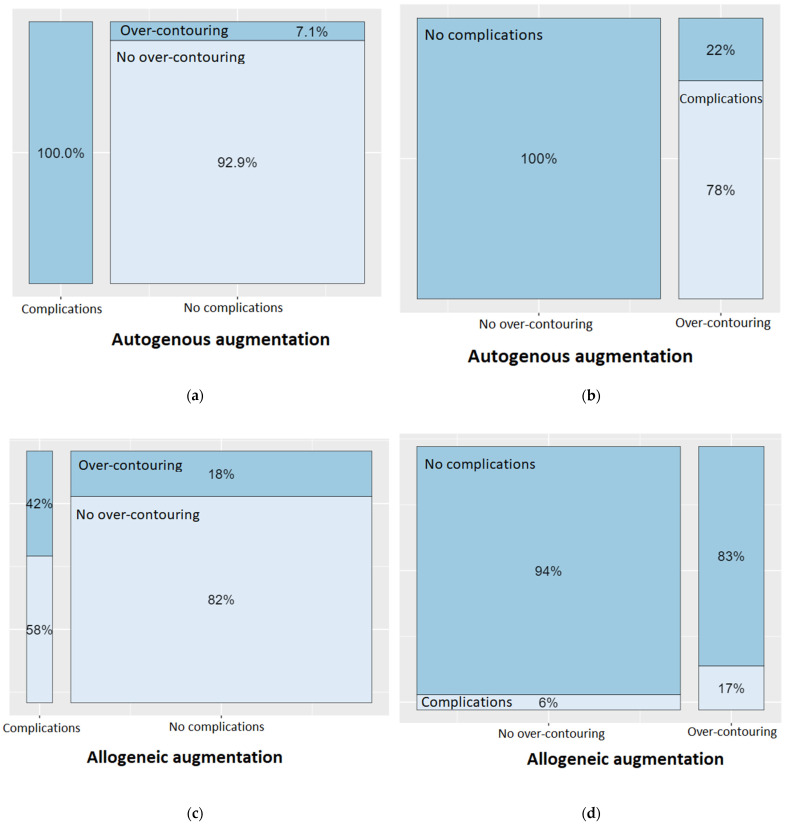
Spineplots on the occurrence of a complication in presence of over-contouring, depending on the type of the bone block (autogenous vs. allogeneic). (**a**) In 100% of complications with autogenous bone blocks, over-contouring was present. (**b**) However, when there was no over-contouring when using autogenous bone blocks, there were no complications at all. (**c**) In 42% of complications with allogeneic bone blocks, over-contouring was present. (**d**) However, when there was no over-contouring when using allogeneic bone blocks, complications occurred in only 6% of cases.

**Figure 6 jcm-12-00006-f006:**
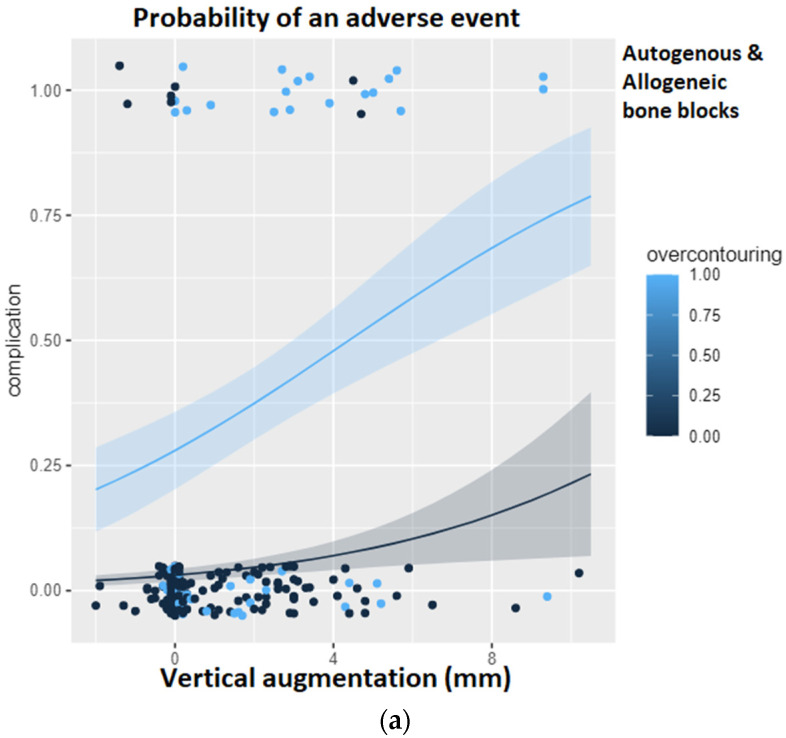
Logistic regression analysis for the probability of a complication given the height of the vertical augmentation and the presence of over-contouring. The light blue curve represents over-contouring, while the black curve represents a perfectly matching augmentation. (**a**) Logistic regression in the overall group. (**b**) Logistic regression restricted to the study group using autogenous bone blocks. It can be seen that with increasing vertical augmentation, the risk of a complication does not increase at all as long as the augmentation remains below the level of the surrounding, natural alveolar ridge. (**c**) Logistic regression restricted to the study group using allogeneic bone blocks.

**Table 1 jcm-12-00006-t001:** Demographic characteristics of the patient groups.

	Autogenous Bone	Allogeneic Bone	*p*-Value
Number of cases	70	151	
Gender	22 males; 48 females	49 males; 102 females	0.504
Age (mean ± std. dev.)	45.2 ± 12.4	51.4 ± 14.2	0.002
Follow-up, years	3.5 ± 1.9	3.5 ± 1.6	0.881
Source of bone material	Iliac crest: *n* = 2 Jaw angle: *n* = 63 Tuber maxillae: *n* = 5	Freeze-dried bone block	
Defect classification	Type II: *n* = 50 (71.4%) Type III: *n* = 14 (20%) Type IV: *n* = 6 (8.6%)	Type II: *n* = 81 (53.6%) Type III: *n* = 50 (33.1%) Type IV: *n* = 20 (13.2%)	0.043
Defect size	Single-tooth gap: 41 (58.6%) Several missing teeth: 29	Single-tooth gap: 55 (36.4%) Several missing teeth: 96	0.002
Location	Maxilla: *n* = 34 (48.6%) Mandible: *n* = 36 (51.4%)	Maxilla: *n* = 98 (64.9%) Mandible: *n* = 53 (35.1%)	0.027
Smoking	*n* = 12 (17.14%)	*n* = 32 (21.19%)	0.305
Medication	Metformin: *n* = 2 Bisphosphonates: *n* = 0 Anticoagulants: *n* = 0	Metformin: *n* = 1 Bisphosphonates: *n* = 7 Anticoagulants: *n* = 1	0.141
Patient compliance	Poor: *n* = 0 Fair: *n* = 3 Good: *n* = 67	Poor: *n* = 2 Fair: *n* = 21 Good: *n* = 128	0.059
Alveolar ridge, height	11.3 ± 2.6	10.1 ± 3.0	0.003
Gain in height (mm)	1.1 ± 1.9	1.3 ± 2.2	0.663
Alveolar ridge, width	1.3 ± 1.0	1.7 ± 1.5	0.501
Gain in width (mm)	4.8 ± 1.2	4.6 ± 1.6	0.467
Over-contouring	*n* = 18 (25.7%)	*n* = 30 (19.9%)	0.381
Implant diameter (mm; median and range)	4.1 (3.3–4.8)	4.1 (2.9–4.8)	0.198
Implant length (mm; median and range)	10.0 (6.6–12.0)	9.0 (6.0–12.0)	0.001

**Table 2 jcm-12-00006-t002:** Complications observed in the two study groups.

	Autogenous Bone	Allogeneic Bone	*p*-Value
Number of cases	70	151	
Wound dehiscence	*n* = 7 (10.0%)	*n* = 8 (5.3%)	0.250
Partial loss of bone block	*n* = 13 (18.6%)	*n* = 7 (4.6%)	0.001
Total loss of bone block	*n* = 1 (1.4%)	*n* = 3 (2.0%)	0.622
Infection	*n* = 0 (0.0%)	*n* = 1 (0.7%)	0.999
Loss of implant	*n* = 0 (0.0%)	*n* = 1 (0.7%)	0.999
Nerve constraint, temporary	*n* = 1 (1.4%)	*n* = 0 (0.0%)	0.317
Number of cases without any complication	*n* = 56 (80.0%)	*n* = 139 (92.1%)	0.013

**Table 3 jcm-12-00006-t003:** Association of potential risk factors with the occurrence of complications after alveolar ridge augmentation.

Variable	Manifestation	Complications Occurred (*n* = 27)	No Complication (*n* = 194)	*p*-Value
Grafting	Autogenous (*n* = 70)	14	56	0.013
material	Allogeneic (*n* = 151)	12	139	
Upper or lower	Maxilla (*n* = 132)	14	118	0.530
jaw	Mandible (*n* = 89)	12	77	
Gender	Male (*n* = 71)	8	63	0.999
	Female (*n* = 150)	18	132	
Age		42.5 ± 11.9	50.3 ± 13.9	0.007
Smoking	Non-smoker (*n* = 177)	16	161	0.018
	Smoker (*n* = 44)	10	34	
Medication	None (*n* = 210)	25	185	0.901
	Anticoagulants (*n* = 1)	0	1	
	Bisphosphonates (*n* = 7)	1	6	
	Metformin (*n* = 1)	0	3	
Patient	Poor (*n* = 2)	2	0	0.001
compliance	Fair (*n* = 24)	3	21	
	Good (*n* = 195)	21	174	
Over-	No (*n* = 173)	7	166	<0.001
contouring	Yes (*n* = 48)	19	29	
ITI defect	Type II (*n* = 131)	9	122	0.024
classification	Type III (*n* = 64)	12	52	
	Type IV (*n* = 26)	5	21	
Defect size	Single-tooth gap (*n* = 96)	10	86	0.676
	Several missing teeth (*n* = 125)	16	109	
Vertical augmentation	Gain in height (mm)	2.9 ± 2.9	1.0 ± 1.9	<0.001
Vertical	<2.55 mm (*n* = 170)	11	159	<0.001
augmentation	≥2.55 mm (*n* = 51)	15	36	
Horizontal augmentation	Gain in width (mm)	4.9 ± 2.0	4.7 ± 1.5	0.456
Bone grafting	Cerabone (*n* = 128)	13	115	0.377
material	None (*n* = 68)	8	60	
	Endobon (*n* = 25)	5	20	

**Table 4 jcm-12-00006-t004:** Multiple logistic regression with the occurrence of a complication as dependent variable and the bone block material (autogenous or allogeneic), smoking (yes or no), vertical augmentation (below or above threshold), and over-contouring (yes or no) as independent variables. Nagelkerke pseudo-R-square: 0.453.

Variable	Odds Ratio [95% Confidence Interval]	*p*-Value
Bone block material—autogenous	3.2 [1.1–9.1]	0.027
Smoking—yes	4.8 [1.2–14.8]	0.007
Vertical augmentation—above threshold	5.0 [1.7–14.1]	0.002
Over-contouring—yes	15.3 [5.2–44.5]	<0.001

**Table 5 jcm-12-00006-t005:** Classification table.

Observed Occurrence of Complications	Predicted Occurrence of Complications		
	No Complication	Complication	Percentage Correct
No complication	193	2	99.0
Complication	13	13	50.0
	Overall percentage		93.2

**Table 6 jcm-12-00006-t006:** Association of smoking with patient compliance.

	Patient Compliance			
	Poor	Fair	Good	Total
Non-smoker	0	16	161	177
Smoker	2	8	34	44
Total	2	24	195	221

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
