# Peer review of "Risk Factors for Complications Following Staged Alveolar Ridge Augmentation and Dental Implantation: A Retrospective Evaluation of 151 Cases with Allogeneic and 70 Cases with Autogenous Bone Blocks"

_jcm, 2022, doi:10.3390/jcm12010006_

Round 1
Reviewer 1 Report (Previous Reviewer 2)
Nice revision
Reviewer 2 Report (Previous Reviewer 1)
well written - interesting paper
This manuscript is a resubmission of an earlier submission. The following is a list of the peer review reports and author responses from that submission.
Round 1
Reviewer 1 Report
Well written paper- relevant content
Please be Constisten throughout the text with Class 3
2.8 Page 4 bottom
Risk of Bias and Discussion: Please be aware of the fact that all surgeries were performed by one surgeon and state the potential risk of bias in regards to the surgeons experience.
Reviewer 2 Report
A interested paper about the rehabilitation of bone defects. However, there are some issues to be revised:
1) The authors should avoid first tense
2) The authors should comment about the different sample size of autogenous bone blocks and the allogenous bone, was a factor influencing the results?
3) Report the kind of sutures used
4)There are several "Error messages" about the reference. The authors should correct them
5) Does the different clinical situation of the patients have a major impact to the selection of the autogenous or allogenous bone blocks? The authors should comment on that.
6) Avoid expression like "on the other hand"
7) All the clinical cases were indicated for single sutures?
8)Since you mention p values in the manuscript, the authors should consider removing from Table 1,2,3,4,5
9) The authors should consider making a composite Figure for Figure 5.